# Leg Dominance as a Risk Factor for Lower-Limb Injuries in Downhill Skiers—A Pilot Study into Possible Mechanisms

**DOI:** 10.3390/ijerph16183399

**Published:** 2019-09-13

**Authors:** Arunee Promsri, Alessia Longo, Thomas Haid, Aude-Clémence M. Doix, Peter Federolf

**Affiliations:** 1Department of Sport Science, University of Innsbruck, 6020 Innsbruck, Austria; arunee.pr@up.ac.th (A.P.); a.longo@donders.ru.nl (A.L.); thomas.haid@uibk.ac.at (T.H.); aude-clemence.doix@uibk.ac.at (A.-C.M.D.); 2Department of Physical Therapy, University of Phayao, Phayao 56000, Thailand; 3Donders Institute for Brain, Cognition and Behaviour, Radboud University, 6525 Nijmegen, The Netherlands

**Keywords:** leg dominance, injury, risk factor, anterior cruciate ligament ACL, balance, neuromuscular control, knee muscle strength, sex difference, kinematic principal component analysis PCA

## Abstract

Leg dominance has been reported as one potential risk factor for lower-limb injuries in recreational downhill skiers. The current study proposed and tested two possible mechanisms for a leg dominance effect on skiing injuries—imbalance of the knee muscle strength and bilateral asymmetry in sensorimotor control. We hypothesized that the knee muscle strength (Hypothesis 1; H1) or postural control (Hypothesis 2; H2) would be affected by leg dominance. Fifteen well-experienced recreational downhill skiers (aged 24.3 ± 3.2 years) participated in this study. Isometric knee flexor/extensor muscle strength was tested using a dynamometer. Postural control was explored by using a kinematic principal component analysis (PCA) to determine the coordination structure and control of three-dimensional unipedal balancing movements while wearing ski equipment on firm and soft standing surfaces. Only H2 was supported when balancing on the firm surface, revealing that when shifting body weight over the nondominant leg, skiers significantly changed the coordination structure (*p* < 0.006) and the control (*p* < 0.004) of the lifted-leg movements. Based on the current findings, bilateral asymmetry in sensorimotor control rather than asymmetry in strength seems a more likely mechanism for the previously reported effect of leg dominance on lower-limb injury risk in recreational downhill skiers.

## 1. Introduction

Downhill or alpine skiing represents one of the most popular recreational winter sports worldwide with a high-risk for injury. Lower-limb injuries are the most common, especially injuries to the knee [1,2,3]. Around half of all serious knee injuries are knee sprains, e.g., of the anterior cruciate ligament (ACL) [4]. The ACL injury risk is three times greater in female than in male skiers [5]. Furthermore, leg dominance, which is defined as the preferential use of one leg over another in performing lower-limb motor tasks [6], has been listed as one of the risk factors for knee injuries. Particularly in female skiers, several previous studies reported a greater injury rate in the nondominant leg compared to the dominant leg [7,8,9]. However, the underlying mechanisms explaining an asymmetry in the injury risk due to leg dominance remain unclear. The current study proposed and tested two possible mechanisms that might lead to an effect of leg dominance on the injury risk: (1) an imbalance of muscle strength and (2) asymmetry in sensorimotor processing which could lead to differences in postural stability or in the ability to avoid excessive mechanical loads.

Muscle strength is one of the key factors for successful sports performance. During skiing, forceful eccentric contractions are required, especially for repeated high-velocity turns [10,11]. Imbalance of the knee flexor and extensor muscle strength is of interest and has been suggested as one possible mechanism for lower-limb injuries [12]. Specifically, if the quadricep muscles generate substantially larger forces as compared to the hamstrings (“quadriceps dominance”), excessive anterior translation may occur during dynamic activities and causing high strain in the ACL. Or the other way around, if the hamstrings are too weak to counteract quadricep forces, the ACL may be at risk of injury [13,14]. Hence, in the current study, isometric knee flexor/extensor muscle strength was tested. It was hypothesized that interlimb differences in the knee muscle strength or in hamstring/quadricep strength ratios would be observable if strength differences or strength ratio differences between legs are a relevant mechanism for an effect of leg dominance on injury rates. Some studies report that quadricep dominance is more prevalent in women [15,16,17,18].

The movements of the left and right extremities are controlled by the right and left brain hemispheres, respectively. Another mechanism explaining the observed difference in injury rates between dominant and nondominant leg might therefore be a difference in the neuromuscular control between the dominant and nondominant leg. Such bilateral asymmetry in neuromuscular control was recently observed in the postural movements during one-leg standing [19,20]. One-leg standing experiments actually simulate aspects of the skiing situation, considering that shifting one’s body weight onto one leg and stabilizing this position is one of the necessary skills in downhill skiing, especially during turning, where the weight distribution differs substantially between outer and inner leg [21]. However, standard measurements (standing upright) of one-leg balancing do not fully resemble the skiing situation, since there are important differences in the body posture and in other boundary conditions between the skiing posture and normal upright standing. For example, the ski boots force the skier into a bent-knee position and the skis and boots produce markedly different weight distributions and lever arms [22], which might affect balancing. Furthermore, postural stability is permanently challenged in skiing, e.g., due to uneven ground or varying snow conditions. Specifically, falls resulting from changing snow conditions has been reported as one cause of skiing injuries [23,24]. Therefore, the current study also investigated the hypothesis of differences in postural control between dominant and nondominant leg. These differences were analyzed in a skiing specific posture, with skiing equipment, and with and without challenging the postural stability by placing the volunteers on a soft surface. 

In summary, the current study is motivated by empirical results reporting significantly higher injury rates for the nondominant leg than the dominant leg in alpine skiing, particularly among female skiers [7,8,9]. We propose that two underlying mechanisms may lead to this asymmetry in injury rates: first, an asymmetry in the knee muscle strength or in the hamstring/quadricep strength ratios between dominant and nondominant leg; or second, a systematic difference in the neuromuscular control between dominant and nondominant leg. The purpose of the current study was to test these proposed mechanisms by determining in experienced recreational alpine skiers if asymmetries exist in knee muscle strength or strength ratios (H1) or in unipedal postural control (H2). We also tested for a sex effect and an interaction between leg dominance and sex for both knee muscle strength and postural control variables.

## 2. Materials and Methods

### 2.1. Participants 

Fifteen well-experienced recreational downhill skiers (8/7 males/females) provided written informed consent for participation. All participants were physically active young adults with at least five season experiences in downhill skiing (aged 24.3 ± 3.2 years, physical activity: participation in sports 5.1 ± 4.2 h/week, skiing experience 19.1 ± 5.5 seasons). The volunteers self-reported no recent musculoskeletal injuries or neurological disorders within the last two years and no participation in any types of specific balance training within the last one year. Most participants (86.7%) reported their right leg as the dominant leg. Leg dominance was defined as the preferred leg for kicking a ball, since this definition had proven most effective in previous studies when determining interlimb differences in unipedal postural control [19,20]. The dominant leg of all participants was coincidental with their dominant hand determined from the writing hand. Characteristics of the participants and of their skiing equipment [7] are listed in Table 1. The current study protocol was approved by the Board for Ethical Questions in Science of the University of Innsbruck, Austria (Certificate 51_2016).

### 2.2. Measurement Procedures and Data Analysis

#### 2.2.1. Knee Muscle Strength Tests

Isometric knee flexor/extensor muscle strength of both legs was measured using an isokinetic dynamometer (CON-TREX MJ; CMV AG, Zurich, Switzerland). Prior to the strength tests, the participants performed a 10-min warm-up on a cycling ergometer (Daum Electronics, Ergo Bike, 8008 TRS, Fürth, Germany) at a pedaling frequency of 70 rpm with the first 5 min at 1 watt.kg^−1^ and the last 5 min at 2 watts.kg^−1^ [25]. After warm-up, the participants were transferred and secured to the isokinetic dynamometer. The participants were asked to sit on the isokinetic dynamometer and dual crossover straps and a waist strap secured the upper-body to restrain the participants from any movement. The knee joint angle was placed at 110° (full extension being 180°) and the hip joint angle was placed at 100° (full extension being 180°). The axis of the isokinetic dynamometer was aligned with the rotation axis of the knee. The lever arm was fixed on the lower leg, about 2 cm above the lateral malleolus of the ankle [26,27].

The testing protocol consisted of two repetitions of isometric maximal voluntary contractions (MVCs) of knee extension and knee flexion for each leg’s muscle groups (quadricep and hamstring groups) similar to protocols that were described earlier [26,27]. Observed torques were corrected for gravitational contributions. Volunteers were verbally encouraged and the testing order (dominant/nondominant; hamstring/quadricep) was randomized. A rest period of five minutes was respected between each MVC. The strength tests were conducted after completing the postural control tests to prevent muscle fatigue affecting postural control.

To examine the knee muscle strength (H1), three variables were computed for each leg of each participant. The peak torque of the knee flexors and the peak torque of the knee extensors were normalized to the individual body mass [28]. The third dependent variable was the ratio between the normalized hamstring and quadricep torques (i.e., HQ ratio) [27,29]. 

#### 2.2.2. Postural Control Tests—Experimental Procedure

The testing protocol and the analysis procedures for the postural control testing are based on an earlier study [19] but were adapted to a skiing-specific situation. The standing trials were executed on two surfaces: a firm (FS) and a soft surface (SS), which were meant to resemble hard pressed and soft snow. For each support surface, all participants completed two 40 s unipedal balancing trials, one for each leg. The sequence of tests was randomly performed and the rest time between trials was three minutes [19]. During the trials, participants were asked to wear their own ski socks and ski boots, to close the clips of their boots as tight as they usually do when they go skiing, and to step into the ski binding. Participants were instructed to rest their hands on the hips and to flex the hip of the lifted leg at approximately 20° [19]. They were to stand still, avoiding any movements not required for balancing such as scratching, to not touch the stance leg with the lifted leg, and to look straight ahead at a 10 cm diameter target placed at individual eye level on a wall approximately five meters away [19].

#### 2.2.3. Postural Control Tests—Instrumentation

Thirty-nine reflective markers were placed over all body segments and on the ski boots according to the “Plug-In Gait” marker setup (Vicon Motion Systems Ltd., Oxford, UK). The position of these markers was captured by a standard eight-camera motion tracking system (Vicon Bonita B10 cameras with Nexus 2.5 software; Vicon Motion Systems Ltd., Oxford, UK) using a sampling rate of 250 Hz. The firm surface was the level, rigid floor of the lab. The soft surface consisted of a rectangle medium-density foam (50 cm width × 180 cm long × 5 cm height) covered by an exercise mat (Airex corona; 100 cm width × 180 cm long × 1.5 cm height; AIREX^®^, Swiss, Switzerland).

#### 2.2.4. Postural Control Tests—Data Analysis

The data analysis method used in this study was described in detail in a previous publication [19]. All kinematic data processing was conducted in the Matlab program (MathWorks Inc., Natrick, MA, USA). Any gaps in marker trajectories were filled by a PCA-based reconstruction technique [30,31]. Postural control tests were conducted using a novel approach, kinematic principal component analysis (PCA), to explore the composition and control of the whole-body postural movements [32,33]. Two PCA analyses were computed, one for each surface condition. In each PCA analysis, all left-leg trials were mirrored and relabeled such that all input appeared as right-leg trials to provide for a direct comparison between legs [19]. Nine asymmetrical markers placed on the upper arms, lower arms, right scapular, upper thighs, and lower thighs were omitted, and the middle 30 s of all balancing trials was selected for further analysis [19]. 

The 30 analyzed markers (x, y, z) represented 90 spatial coordinates (i.e., 90 dimensional posture vectors). Each trial provided a matrix of 7500 (30 s at 250 Hz) posture vectors. The matrixes of the dominant and nondominant standing trials were centered [30], normalized [30], weighted [34,35], and then concatenated from all participants and all same-surface balancing trials to produce a 225,000 × 90 input matrix for the PCA (250 [sampling rate] * 30 [trial duration] * 15 [number of participants] * 2 [number of trials] × 90 [marker coordinates]). The PCA was calculated by performing a singular value decomposition on the covariance matrix of the input matrix. The PCA yielded a set of PC_k_-vectors, where k represents the order of movement components. Each PC vector defined a specific pattern of correlated marker movements, which we call “*principal movement*” (PM). Animated stick figures can be created from each individual PC_k_-vector to produce a visual representation for each PM (Figure 1). In postural control trials, the PMs closely resemble the different postural movement strategies, such as the ankle or hip strategy [36]. The PCA also yielded a set of eigenvalues and scores. The scores are obtained by projecting the normalized posture vectors onto the PC_k_ vectors and thus represent the actual postural movements expressed in the coordinates defined by the PC_k_ vector basis, i.e., in PM coordinates. The eigenvalues quantify the variance represented in each PM; they can be expressed as relative variance, i.e., the percentage of the total postural variance that is represented in each PM. 

In analogy with Newton’s mechanics, the resultant principal component scores, i.e., PC_k_ scores, can be interpreted as “principal” positions PP_k_(t). If the PP_k_(t) are differentiated twice, one obtains “principal” accelerations PA_k_(t); where (t) indicates that these variables are functions of time t. To avoid noise amplification in the differentiation processes [37], a Fourier analysis was conducted on the raw PP_k_(t), which revealed that the highest power resided in frequencies around 2–4 Hz, but visible power was still found in the frequency range between 5 to 8 Hz. The PP_k_(t) were, therefore, filtered with a third-order zero-phase 8 Hz low-pass Butterworth filter before the differentiation.

To investigate postural control (H2), two main types of PCA-based variables were determined. First, the coordinative structure was characterized by assessing the composition of the single-leg balancing postural movements. Thereto, the participant-specific relative explained variance, *PP*_*rVAR*_k_ were calculated from the PP_k_(t). The *PP*_*rVAR*_k_ quantify how much (in percent) each individual PM contributed to total variance in postural positions [19,32,36]. Differences in *PP*_*rVAR*_k_, for example, between balancing on the dominant leg and the nondominant leg, would therefore indicate a difference in the coordination structure of the overall postural movements. Second, the control of movement components was explored by computing two participant-specific PA variables; the relative explained variance of the PA_k_(t), *PA*_*rVAR*_k_, which quantify how much (in percent) individual PM accelerations contributed to total variance in postural accelerations [38], and the number of zero-crossings (*N*_k_), which quantifies how often the PM acceleration changes direction [19,20,39,40]. Differences in *PA*_*rVAR*_k_ between conditions indicate a difference in the coordination of postural accelerations. Differences in *N*_k_ can be interpreted as differences in how tightly a movement component is controlled [19,20,39,40]. In this study, the first eight PMs were considered (k = 1…8, for *PP*_*rVAR*_k_, *PA*_*rVAR*_k_, and *N*_k_) since the underlying PC_k_ vectors proved to be robust in a cross-validation test [39].

### 2.3. Statistical Analysis

All statistical analyses were performed using the SPSS software (IBM SPSS Statistics 24, SPSS Inc., Chicago, IL, USA) with the alpha level set at α = 0.05. Shapiro–Wilk tests were used to test for normal distribution of the considered variables. Independent sample *t*-tests were used to test for differences in characteristics between male and female skiers. Main and interaction effects of leg dominance (within-subjects) and sex (between-subjects) on muscle strength and postural control variables were tested with a split-plot repeated measures ANOVA for all variables separately. Partial eta squared values were used to report the effect size. Based on the PCA analysis, individual principal movements (PMs) represent different postural movement strategies; hence, one ANOVA was tested for each movement component (PM_1–8_) of each variable (*PP_rVAR*, *PA_rVAR*, and *N*). The variable *rVAR* was not normally distributed in some movement components (for firm surface; *PP*_*rVAR*_4–8_ and *PA_rVAR*_6–8_, for soft surface; *PP_rVAR*_2,6–8_ and *PA_rVAR*_4_,_7_). In these cases, the corresponding nonparametric tests (Wilcoxon for leg preference; Mann–Whitney for sex effects) were conducted and Rosenthal’s r was used as effect size. Post-hoc observed power (1-β) was calculated using the software G*Power (Version 3.1.9.2; University of Kiel, Kiel, Germany) and, for nonsignificant results, the beta level was set to β = 0.2, corresponding to a threshold of 0.8 for the observed power. For each PCA variable, the Bonferroni–Holm correction [41] was used to adjust the rejection criteria for multiple comparisons among the eight PMs, thus controlling for the family-wise error rate.

## 3. Results

### 3.1. Knee Muscle Strength 

As shown in Table 2, no leg dominance effect was observed in all muscle strength variables, while only one sex effect was found in the normalized hamstring torque (F_(1, 13)_ = 4.83, *p* = 0.047, η_p_^2^ = 0.271). In addition, no interaction between leg dominance and sex was found in all muscle strength variables.

### 3.2. Postural Control 

All participants were able to complete all balancing trials without touching the floor by the lifted leg to regain balance. A description of the main movement characteristics and a visualization of the first eight movement components (PM_1–8_) are represented in Table 3 and Figure 1, respectively. The first eight PMs together explained 91.4% and 88.3% of the total postural variance on the firm and on the soft ground, respectively.

#### 3.2.1. Composition of Single-Leg Balancing Movements

Significant leg dominance effects were observed in the composition of the postural movements (*PP_rVAR_k_*) when balancing on the firm surface (Figure 2A: first row). Specifically, balancing on the dominant leg showed greater proportions in two movement components, PM_3_ (*PP_rVAR*_3_; F_(1, 13)_ = 10.84, *p* = 0.006, η_p_^2^ = 0.455) and PM_6_ (*PP_rVAR*_6_; Z = 3.12, *p* = 0.002, r = 0.806), representing predominantly hip movements of the lifted leg (Table 3 and Figure 1). The observed power (1-β) exceeded 0.8 only for the movement components where significant differences were found (PM_3_ and PM_6_). All nonsignificant results had an observed power between 0.05 and 0.709. 

No significant sex difference was found when balancing on both firm (Figure 2A: second row) and soft (Figure 2B: second row) surfaces when applying the Holm–Bonferroni correction. However, when balancing on FS (Figure 2A: second row), two movement components suggested a tendency for a sex effect: female skiers showed an increased anteroposterior sway around the knee joint as seen in PM_1_ (*PP_rVAR*_1_; F_(1, 13)_ = 5.63, *p* = 0.034, η_p_^2^ = 0.302), while male skiers showed an increased hip flexion/extension of the lifted leg as seen in PM_3_ (*PP_rVAR*_3_; F_(1, 13)_ = 4.76, *p* = 0.042, η_p_^2^ = 0.280). Observed power was smaller than 0.55 in all tests. No interaction between leg dominance and sex was found. 

#### 3.2.2. Control of Movement Components 

Significant differences in the composition of postural accelerations (*PA_rVAR_k_*) were observed between dominant and nondominant leg only when balancing on the firm (Figure 3A: first row), but not on the soft surface (Figure 3B: first row). Specifically, balancing on the dominant leg showed greater contributions to the overall acceleration by two movement components representing predominantly hip movements of the lifted leg: in PM_1_ (*PA_rVAR*_3_; F_(1, 13)_ = 11.91, *p* = 0.004, η_p_^2^ = 0.478, 1-β = 0.89) and in PM_6_ (*PA_rVAR*_6_; Z = 3.40, *p* = 0.001, r = 0.880, 1-β = 0.96). In the other movement components, no significant differences were found with observed statistical power smaller than 0.60 in all tests. No sex effect (Figure 3: second row) or interaction between leg dominance and sex was found.

No significant difference in *N_k_* was found for leg dominance or sex effects, neither when balancing on firm (Figure 4A), nor when balancing on the soft (Figure 4B) surface. However, when unipedally balancing on the firm surface, the dominant leg showed a tendency of increased *N_k_* compared to the non-dominant leg in two movement components, k = 3 and k = 7, characterizing predominantly hip flexion/extension of the lifted leg (*N*_3_; F_(1, 13)_ = 9.14, *p* = 0.010, η_p_^2^ = 0.413, 1-β = 0.80 and *N*_7_; F_(1, 13)_ = 5.81, *p* = 0.031, η_p_^2^ = 0.309, 1-β = 0.61). For all other movement components, observed power did not exceed 0.44. No interaction between leg dominance and sex was found.

## 4. Discussion

The current study proposed and tested in recreational downhill skiers two possible mechanisms for an asymmetry in knee injury risk based on leg dominance—an imbalance of the knee muscle strength and interlimb differences in sensorimotor control. As a secondary goal, we tested for sex effects and interactions between sex and leg dominance. The findings highlighted that the knee muscle strength was symmetrical between the dominant and nondominant legs. Hence, Hypothesis H1 was not supported. We found differences in normalized hamstring torque between the sexes, but no other differences and no interaction effects. In contrast, clear differences in sensorimotor control due to leg dominance, particularly in variables characterizing the coordination of postural movements, were found (Hypothesis H2 was supported). Variables characterizing how tightly movement components were controlled also showed a tendency, but were not significant when controlling for the family-wise error. However, the motor control differences due to leg dominance were observed only when skiers unipedally stood on the firm surface. On both surfaces, no significant sex effects or interaction effects were found in the variables characterizing unipedal postural control.

### 4.1. Knee Muscle Strength 

Regarding the first hypothesis, the current findings illustrated that the knee muscle strength seemed not affected (very small effect sizes) by the preferential use of one leg over another; however, the statistical power was not sufficient to actually reject H1. This finding was in line with previous studies comparing isokinetic knee strength between the dominant and nondominant leg, which reported no interlimb differences in knee muscle strength measured in young healthy athletes [42,43]. One possible reason for the symmetry in contralateral muscle strength might be that all recreational skiers who participated in the current study were physically active young adults whose active participation in sports seemed to have trained both limbs similarly. A notable observation in the current study was that the HQ ratio, representing the balance between hamstring and quadricep muscle strength [44] of both legs were on the lower end of the range reported in the literature (0.5 to 0.8) for healthy participants [12]. This observation could suggest that participants of the current study seemed to have an imbalance between hamstring and quadricep strength in both legs. To prevent knee injury and to improve knee joint stability, improving the hamstring muscle strength would be indicated and concentric HQ should be assessed in future studies examining dominant and nondominant legs. Specifically, a concentric HQ ratio closer to 0.8 might reduce risk of hamstring strain [12]. 

When considering the differences in normalized muscle strength between sexes, our findings agree with previous findings, reporting that male skiers showed greater normalized hamstring torques in both eccentric and concentric contractions than female skiers [29]. The hamstring torque is particularly interesting in the context of prevention of the knee injuries, since the hamstring muscles act as an ACL agonist to resist anterior translation of the tibia relative to the femur [45,46]. As previously reported, female athletes who later suffered an ACL injury often had a combination of decreased hamstring strength, but not decreased quadricep strength, compared to male athletes [47].

### 4.2. Postural Control 

In agreement with the second hypothesis, the dominant (kicking) and nondominant (supporting) leg used different postural movement strategies (*PP_rVAR*) to control unipedal balancing on the firm surface. Specifically, standing on the dominant leg showed greater hip movements of the lifted leg coupled with upper body movements (PM_3_ and PM_6_), suggesting that the skiers utilize the lifted (free) leg in their balancing movements to maintain postural stability. Surprisingly, greater amplitudes of PM accelerations (*PA_rVAR*) were found in these two movement components when balancing on the dominant leg, confirming that skiers move their free leg faster [38] than when balancing on the nondominant leg. The same tendency was also seen from *N*_k_, characterizing how tightly the lifted-leg movements (PM_3_ and PM_7_) were controlled. Interestingly, bilateral asymmetry in unipedal postural control disappeared when balancing on the unstable, soft surface. One interpretation could be that interlimb differences emerged in the surface condition in which leg movements are a major contribution to postural control, as observed when balancing on the firm surface. In contrast, trunk movements were more dominant in the more difficult task of balancing on the soft surface (Table 2). This result might imply that an asymmetry in the injury risk between dominant and nondominant leg may occur particularly when skiing on hard snow rather than on soft snow; however, we are not aware of empirical data that could support this interpretation of our results. 

Male and female skiers trended to use different main motor strategies to achieve the postural control in the balancing trials. The tendency was corroborated with strong effect sizes [48] observed in the anteroposterior sway around the knee joint (PM_1_) and in movements of the lifted leg (PM_3_). Female skiers seemed to control the whole-body stability more through the stance leg, while in male skiers, the role of higher order movement components for postural movements was increased. One potential reason for reduced movements of the free leg in female skiers might be the influence of their ski equipment, which was heavier relatively to their body size compared to males. Another notable phenomenon observed in the current study was that the “ankle strategy”, a whole-body sway around the ankle, which is typically observed in unipedal barefooted balancing [19,36] manifested in the current study as a “knee strategy”, as seen in the first movement component (PM_1_). Apparently, wearing ski boots, where the ankle joint is slightly dorsiflexed and fixed in the ski boot, leads to a shift of the sway axis to the knee joint. The greater utilization of PM_1_ in female skiers should be considered for knee injury prevention because most knee injuries occur due to a twisting fall in which the axis of the whole movement is the knee joint [49]. 

Finally, that interlimb differences in unipedal postural control emerged in specific movement components is in line with previous studies [19,20], and agrees well with the assumption of the minimal intervention principle [50,51]. The minimal intervention principle proposes that the sensorimotor controller (in our case the controller for postural movements) corrects or controls only movement dimensions/components which interfere with task goal; otherwise, it will ignore variability in movement dimensions [50,51,52]. Differences between groups or between dominant/nondominant stance legs seem to emerge particularly in movement components that require a relatively tight control [19,20], and could suggest that the neuromuscular controller either has difficulty to adapt, or that it follows a different control strategy. More research on the specificity of control characteristics in different movement dimensions is needed. 

### 4.3. Limitations 

The first main limitation of the current study was relatively small sample size. The main reason for this small sample was that the current study permitted only well-experienced recreational skiers with at least five season experiences into the study and all participants needed to be free from any musculoskeletal injury or neurological disorders within the last two years. In this context, another limitation is that the information on musculoskeletal injuries or neurological disorders was only self-reported. A physical screening of the volunteers was not performed. Another limitation was that in the current study, only isometric hamstring and quadricep strength was tested. Alpine skiing demands various forms of muscle work; however, there are phases at the end of each turn where relatively high muscle forces have to be produced at almost isometric conditions [53]. 

## 5. Conclusions

The current study proposed and tested two possible injury-related mechanisms due to leg dominance—imbalance of the knee muscle strength and bilateral asymmetry in sensorimotor control. The results revealed that in well-experienced recreational downhill skiers, leg dominance seemed to affect only sensorimotor control—specifically coordination of postural movements, but not knee muscle strength. We suggest that bilateral asymmetry in neuromuscular control could be considered as a potential mechanism for the previously reported asymmetry in the risk for lower-limb injuries between the nondominant and dominant legs. Recreational downhill skiers may want to consider assessing interlimb differences in balancing and motor control skills and may want to adjust training if misbalances are found.

## Figures and Tables

**Figure 1 ijerph-16-03399-f001:**
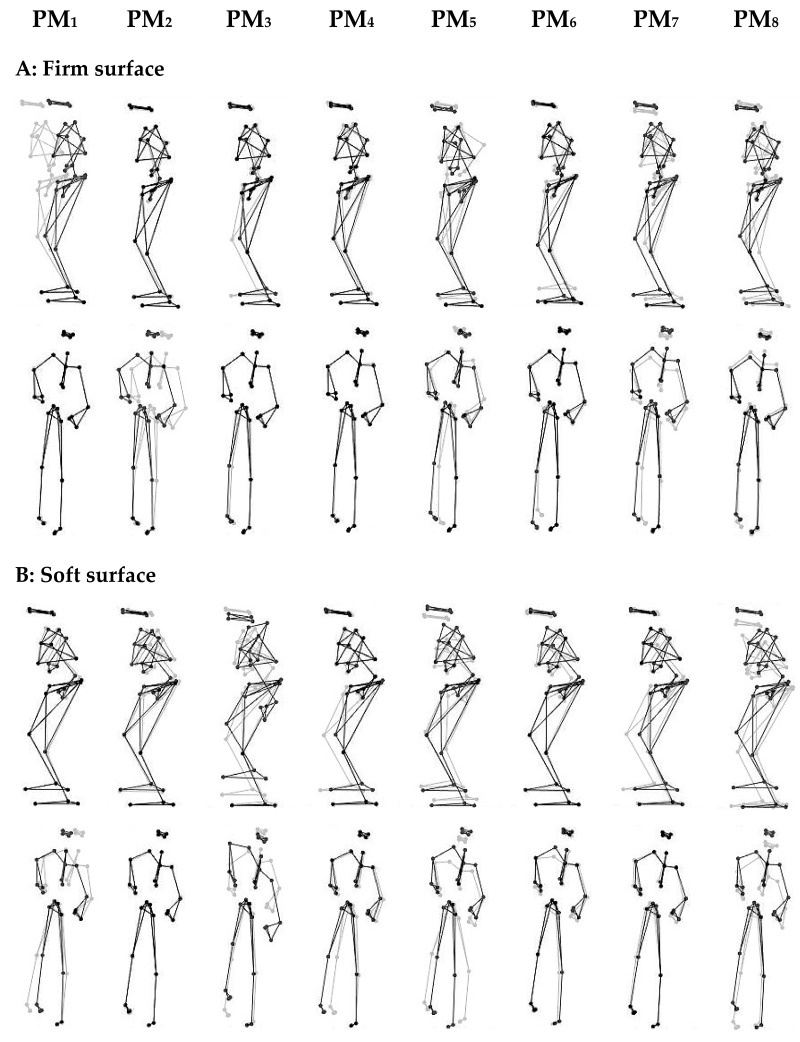
Visualization of the first eight principal movements (PM_1–8_) of balancing on (**A**): firm surface and (**B**): soft surface. Gray and black lines/dots show the extreme posture in opposite directions. For higher-order movement components, the movement amplitude was artificially amplified for a better visualization (Firm surface: amplification 5× for PM_1–4_, and 10× for PM_5–8_; Soft surface: amplification 1× for PM_1–4_, and 2× for PM_5–8_). Note: movements are clearer and can be more easily characterized when viewed in animated stick figure videos: Appendix A for balancing on the firm and soft surfaces, respectively.

**Figure 2 ijerph-16-03399-f002:**
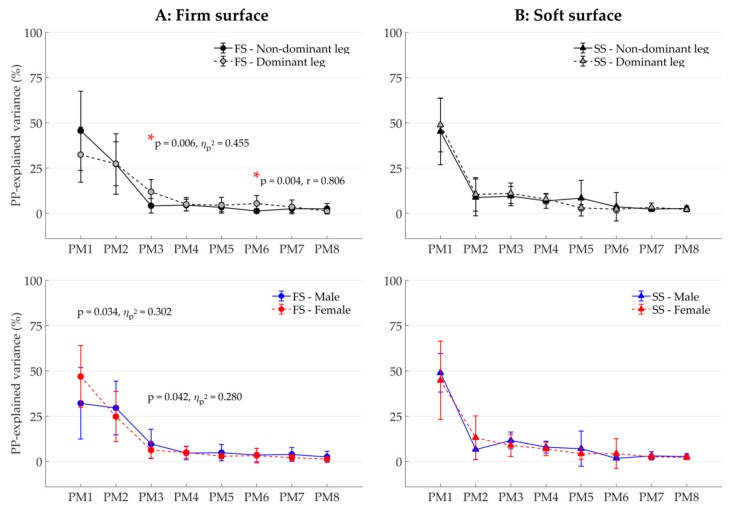
Relative variance spectra of principal positions *PP_rVAR_k_* (mean ± SD) of the first eight principal movements (PM_1–8_) of balancing on the firm surface (FS; column (**A**)) and the soft surface (SS; column (**B**)). Leg dominance effects are examined in the first row and sex effects in the second row. Note: *p*-Values smaller than 0.05 are shown; however, only results satisfying the Bonferroni–Holm criterion [41] were considered significant and marked with the symbol *.

**Figure 3 ijerph-16-03399-f003:**
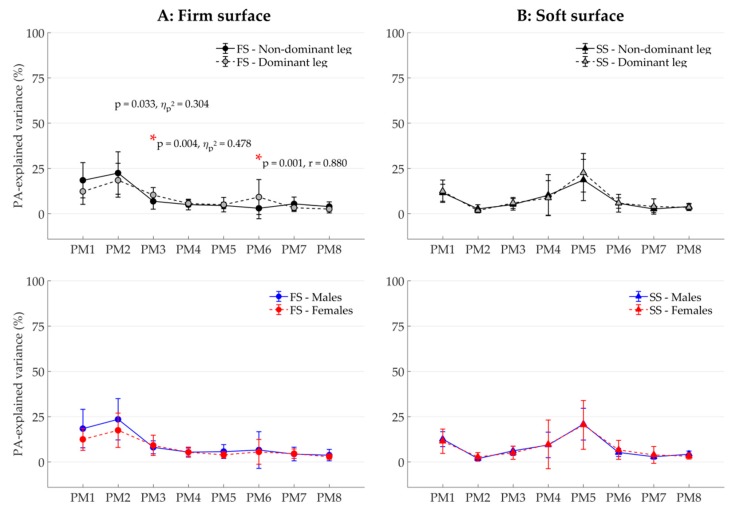
Relative variance spectra of principal accelerations *PA*_*rVAR_k_* (mean ± SD) of the first eight principal movements (PM_1–8_) of balancing on the firm surface (FS; column (**A**)) and the soft surface (SS; column (**B**)), representing the leg dominance effect in the first row and the sex effect in the second row. Note: *p*-Values smaller than 0.05 are shown; however, only results satisfying the Bonferroni–Holm criterion [41] were considered significant and marked with an asterisk *.

**Figure 4 ijerph-16-03399-f004:**
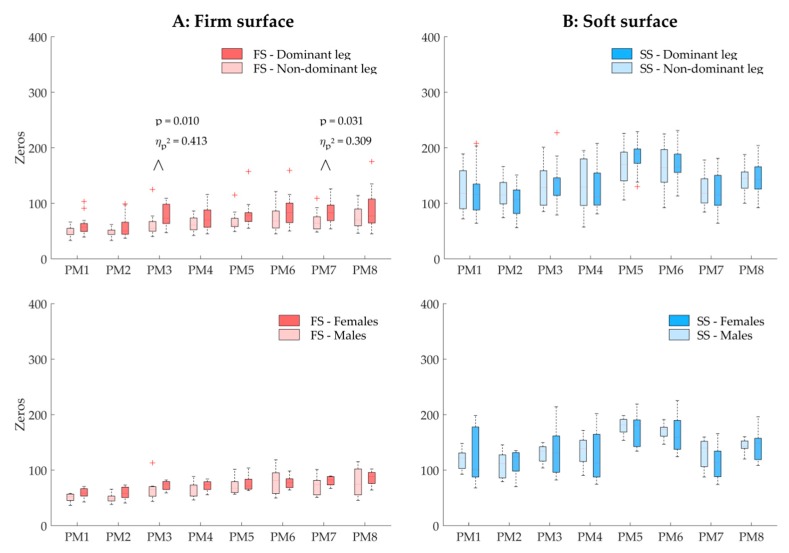
Boxplots of the number of zero-crossings *N_k_* (zeros) of the first eight principal movements (PM_1–8_) of balancing on the firm surface (FS; column (**A**)) and the soft surface (SS; column (**B**)), representing the leg dominance effect in the first row and the sex effect in the second row. Note: *p*-Values smaller than 0.05 are shown; however, none of the differences satisfied the Bonferroni–Holm criterion [41].

**Table 1 ijerph-16-03399-t001:** Characteristics of participants and their ski equipment (mean ± SD). Significant differences at *p* < 0.05 between males and females are symbolized with an asterisk.

	Male (*n* = 8)	Female (*n* = 7)	*p*-Value
Age (year)	24.8 ± 3.3	23.9 ± 3.3	0.611
Weight (kg)	78.9 ± 7.7	59.5 ± 6.4	<0.001 *
Height (m)	181.3 ± 5.2	166.6 ± 4.9	<0.001 *
Body mass index (kg/m^2^)	24.0 ± 2.2	21.4 ± 1.9	0.031 *
Physical activity participation (h/week)	3.9 ± 2.8	6.5 ± 5.3	0.232
Skiing experience (seasons)	18.3 ± 6.8	20.1 ± 3.8	0.529
Skiing ability, *n* (%)			
Carving	8 (100%)	7 (100%)	-
Piste difficulty (Off-piste)	7 (87.5%)	7 (100%)	-
Characteristics of ski equipment			
Ski length (cm)	173.5 ± 10.8	175.9 ± 8.5	0.651
Ski length to body height ratio (%)	95.7 ± 5.8	105.7 ± 7.4	0.011 *
Ski boot weight (kg, per side)	2.3 ± 0.3	2.1 ± 0.3	0.101
Ski weight (kg, per side)	3.5 ± 0.5	3.5 ± 0.5	0.325
Ski equipment weight to body weight ratio (%)	7.4 ± 1.0	9.1 ± 1.7	0.037 *

**Table 2 ijerph-16-03399-t002:** Comparison of the normalized knee muscle strength (Nm/kg; mean ± Std. Deviation) of leg dominance (DO: dominant leg versus ND: nondominant leg) and sex (male versus female) effects and the interaction effects between leg dominance and sex. *Effect size* refers to partial eta squared η_p_² and the observed power is listed as 1-β. Note: the symbol * represents a significant difference at *p* < 0.05.

**Leg Dominance Effects**	**DO (*n* = 15)**	**ND (*n* = 15)**	***p*-Value**	**η_p_²**	**1-β**
Normalized hamstring torque Nm/kg	1.65 ± 0.32	1.63 ± 0.34	0.749	0.008	0.060
Normalized quadricep torque Nm/kg	3.64 ± 0.79	3.63 ± 0.77	0.905	0.001	0.051
HQ ratio	0.46 ± 0.09	0.46 ± 0.09	0.882	0.002	0.053
**Sex Effects**	**Males (*n* = 8)**	**Females (*n* = 7)**	***p*-Value**	**η_p_²**	**1-β**
Normalized hamstring torque Nm/kg	1.79 ± 0.34	1.47 ± 0.17	0.047*	0.271	0.529
Normalized quadricep torque Nm/kg	3.94 ± 0.85	3.30 ± 0.42	0.096	0.199	0.384
HQ ratio	0.46 ± 0.08	0.46± 0.09	0.872	0.002	0.053
**Interaction Effects:**	**Males (*n* = 8)**	**Females (*n* = 7)**	***p*-Value**	**η_p_²**	**1-β**
**Difference between DO and ND**					
in normalized hamstring torque	0.00 ± 0.28	0.04 ± 0.21	0.763	0.007	0.059
in normalized quadricep torque	0.00 ± 0.33	0.03 ± 0.69	0.897	0.001	0.051
in HQ ratio	0.01 ± 0.07	0.00 ± 0.07	0.861	0.002	0.053

**Table 3 ijerph-16-03399-t003:** The relative explained variance of principal positions (*PP_rVAR*; mean ± SD) and of principal accelerations (*PA_rVAR*; mean ± SD) averaged over both dominant and nondominant leg trials, together with a qualitative description of the main movement characteristics of the first eight principal movements (PM_1-8_) on **A:** firm surface and **B:** soft surface.

PM	*PP_rVAR* (%)	*PA_rVAR* (%)	Main Movements
**A:**	**Firm Surface**		
1	39.0 ± 19.7	15.4 ± 9.0	Anteroposterior sway around the knee joint
2	27.3 ± 14.3	20.5 ± 10.6	Mediolateral sway around the ankle joint
3	8.0 ± 6.7	8.6 ± 4.5	Hip flexion/extension of the lifted leg coupled with small rotation of the upper body
4	4.8 ± 3.4	5.4 ± 2.5	Trunk lateral bending
5	3.9 ± 3.5	4.8 ± 3.1	Trunk rotation in the transverse plane
6	3.4 ± 3.8	6.1 ± 8.4	Hip abduction/adduction of the lifted leg
7	3.0 ± 3.1	4.4 ± 3.2	Hip flexion/extension of the lifted leg coupled with lateral trunk bending
8	1.9 ± 2.3	3.3 ± 2.4	Pelvic rotation coupled with hip flexion/extension
**B:**	**Soft Surface**		
1	47.0 ± 16.5	12.0 ± 5.5	Mixed trunk lateral bending and rotation coupled with hip abduction/adduction of the lifted leg
2	9.6 ± 9.6	2.3 ± 1.8	Anteroposterior sway around the knee joint
3	10.3 ± 5.5	5.5 ± 3.1	Mediolateral sway around the hip joint
4	7.4 ± 3.5	9.5 ± 10.5	Hip abduction/adduction of the lifted leg coupled with hip flexion/extension
5	5.7 ± 7.4	20.7 ± 11.2	Trunk flexion/extension
6	2.9 ± 5.7	5.9 ± 4.0	Hip flexion/extension of the lifted leg
7	2.8 ± 1.9	3.4 ± 3.4	Small trunk rotation
8	2.4 ± 1.3	3.7 ± 1.7	Moving up and down of the whole body

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
