# Peer review of "Leg Dominance as a Risk Factor for Lower-Limb Injuries in Downhill Skiers—A Pilot Study into Possible Mechanisms"

_ijerph, 2019, doi:10.3390/ijerph16183399_

Round 1
Reviewer 1 Report
The present study proposed and tested two possible mechanisms for a leg dominance effect on the risk factor for skiing injuries – the imbalance of the knee muscle strength and the bilateral asymmetry in sensorimotor control. However, despite this manuscript presenting clinical relevance for the injury prevention and rehabilitation fields, as well as adequacy of the scope of the Journal, it still needs improvement, which is considered as a prerequisite to a future publication.
Firstly, I think that the objectives of the study are very broad. The study design and the methodological conditions do not allow it to extrapolate its results to such great magnitude. Thus, authors should exercise caution in commenting on results and related clinical impacts, as the sample size is small for large statements. Although the authors have discussed sample size as one of the limitations of the study, a sample power analysis may be useful to better substantiate the significance of the results. The authors used Cohen's d-index to elucidate the differences found from the clinical point of view, but the sample size calculation seems valid to avoid bias in interpreting results.
Another relevant methodological point, which needs to be better considered and discussed, is the use of an isometric torque assessment. It seems that the dynamometer used allows the dynamic torque evaluation. So, it is very important that the authors adequately justify this aspect when addressing the limitations of the study.
Other considerations are made the following specific evaluation.
TITLE is presented in a clear and concise form, which is consistent to the authors investigation. The authors launch a startling question that draws attention to the reader.
The authors use “one-leg stance”; “unstable surface”; “sex difference”; “movement component”; “principal component analysis PCA”; “minimal intervention principle” as keywords. I think that the descriptors used don’t seem appropriate. It was suggested the inclusion of the more generic descriptors as “ANTERIOR CRUCIATE LIGAMENT” (which is one of most important themes of the study); “INJURY”; “RISK FACTORS”; “KINEMATICS”, “KINETICS” and “BALANCE””.
The ABSTRACT includes a brief background and the purpose of the study (to test two possible mechanisms for a leg dominance effect on skiing injuries – imbalance of the knee muscle strength and bilateral asymmetry in sensorimotor control). The description of the METHODS should be slightly expanded to better explain the procedures. The RESULTS (lines 23-25) should be improved in order to better highlight the importance of the study, with quantitative results that summarize the article's main findings. The CONCLUSION is directly related to the objectives.
LINE 16 – I suggest modify “muscle strengths” to “muscle strength” both in the ABSTRACT and in the main text.
LINE 19 – Please add the unit “years”
LINE 20 – Please specify the equipment utilized for isometric assessment. This information is relevant to the ABSTRACT.
LINE 21 - Please specify the kinematic evaluation mode (2D or 3D). This information is relevant to the ABSTRACT.
LINE 27 – I doubt if it is really necessary to describe the term “empirically observed” in the ABSTRACT, since the context involved is not fully explained at this point in the manuscript
INTRODUCTION
The authors briefly point out possible risk factors for ACL injuries in downhill or alpine skiing, emphasizing the biomechanical and neuromuscular factors related do leg dominance. I think the objectives is well written, and that the hypothesis is clearly.
Authors mention the “imbalance of muscle strength” as one of two possible mechanisms that might lead to an effect of leg dominance on the ACL injury risk. I think it would be more interesting to use the term "imbalance of thigh muscle strength" so that it is clear that the study will only address the evaluation of thigh muscles torque (quadriceps and hamstrings). Thus, the reader will already be aware that no proximal (hip muscles) or distal (ankle muscles) investigation will be performed.
The authors problematize the study and defined the purpose of the work and its significance, including specific hypotheses being tested. I think the authors could also use the concept of “quadriceps dominance” to explain why women are at a higher risk of ACL injury compared to men. Although this theme is not being directly evaluated in the present study, women appear to preferentially use the quadriceps more than males in order to stiffen and stabilize the knee joint. In addition, females tend to land from a jump with less knee flexion than males. This stress in combination with other lower limb biomechanical disorders, increases the risk of ligament injury.
I suggest modify “muscle strengths” to “muscle strength” both in all the main text.
I believe that these alterations could produce a more complete text to the reader who has an interest in the area, and to reinforce the importance and rationale for the study.
LINE 45 – I suggest modify to “could lead to differences in postural stability”
LINE 49 – It is suggested modify “Imbalance of the knee muscle strengths, particularly an imbalance between the knee flexors and extensors,” to “Imbalance of the knee flexors and extensors muscles strength”.
LINES 52-53 - Please rewrite the sentence “and the ACL will experience higher-than-normal strains”.
LINES 54-55 - I do not think this sentence is necessary as a justification of the methods to be used.
LINES 55-56 – Please standardize throughout the text: knee flexors/extensors; knee extensors/flexors; quadriceps/hamstrings or hamstrings/quadriceps. It is common to use knee flexors/extensors ratio; or hamstrings/quadriceps ratio.
LINES 67-73 – In this moment there are a number of unreferenced concepts and statements. It is suggested to the authors to base these concepts in the literature.
LINES 73-75 – Again, I do not think this sentence is necessary as a justification of the methods to be used.
LINES 84-85 – I think the second purpose (to test for a sex effect and an interaction between leg dominance and sex for both, knee muscle strength and postural control variables) seems difficult to contemplate with such a small sample. Please, review it.
METHODS
In the most part, the methods and procedures for data collection are described clearly and seem appropriate, in sufficient detail to allow others to replicate and build on published results. The methods comply with ethical principles. The authors pointed out the favorable opinion of the Ethics Committee. I suggest describing the procedures in the order in which they were performed, so first the balance and postural control assessment, and then the isometric torque assessment. Is it an own methodology or based on previous studies? Please report it. The analyses seem in most part appropriate to meet the objectives of the study. However, this section is unnecessarily long, especially in the kinematic analysis. Please shorten if possible. Specific considerations are followed.
LINE 88 - Did the authors have a sample size calculation? Please report it.
LINE 89 – It would be interesting to define the concept of “physically active” or adopt a literature referential for this condition.
LINE 91 – It is suggested to review the sample average time of skiing experience, as it seems not consistent with the data in the table.
LINES 91-92 – “The volunteers self-reported no recent musculoskeletal injuries or neurological disorders within the last two years”. Authors do not perform physical screening to check for inclusion and exclusion criteria? This information is extremely relevant because it is directly related to the criteria adopted by authors for the final sample composition. If any pre-existing conditions exist but they were not reported by volunteers, could the study results be different? It is suggested to the authors to review this aspect and make due consideration in the limitations of the study, if necessary.
TABLE 1 – Verify the skiing experience data, as it seems not consistent with the data in the main text.
LINE 103 – Why the authors perform an isometric torque assessment when an isokinetic dynamometer was available? Please justify use or make due consideration in the limitations of the study, if necessary.
LINE 114 – Authors should mention the use of gravitational correction measures.
LINE 115 – Please replace “hamstring” to "hamstrings" throughout the text.
LINES 116-117 - Please refer to the protocol used for isometric evaluation.
Is there any previous study used the same protocol? This is important information regarding the reproducibility of the study.
LINE 115-119 – In the same line, I really miss some relevant information about the isometric assessment protocol used. Was there any familiarization protocol? Was there any encouragement verbal command during the tests? Which limb was tested first (D or ND)? Which muscle group was tested first? These are important information to the reproducibility of the study. Some of these precautions were taken in the description of the postural control assessment.
LINES 121-123 – Please rewrite the sentence to make it clearer.
LINES 130-133– Please refer the experimental procedure for postural control tests. This is important information regarding the reproducibility of the study.
LINES 150-152 – Please refer the experimental procedure for postural control tests. This is important information regarding the reproducibility of the study.
LINE 162 – “Animated sick figures” or “Animated stick figures”?
RESULTS
The main results are pointed out in a concise and precise form, according to the norms of the journal. There is no duplication of information in charts, tables and text. The figures (04) and tables used (03) include the requirements necessary to address the main issues. However, perhaps the number of figures can be reduced.
Table 2 – Please report the units of measurement.
Figure 1 – Interesting figure, but it can be removed (if necessary) without damaging the manuscript, since authors recognize that movements are clearer and can be more easily characterized when viewed in animated stick figure videos.
DISCUSSION
The results verified in the present manuscript are at mostly confronted with other previous works. In most part, authors pointed convergences and divergences between the results, and the interpretations of the findings demonstrate ownership in relation to the central theme of the article. Some studies of a similar nature are cited to assist in the foundation of authors’ considerations. However, it is emphasized that the discussion still needs improvements in some key points, as addressed below.
I suggest modify “muscle strengths” to “muscle strength” both in all the main text.
LINES 304-316 – I believe that the authors must consider the methodological and clinical differences between isometric and dynamic torque assessments in the interpretation of results and in confrontation between the studies cited. There is a very robust body of knowledge behind it, which needs to be described.
LINES 308-309 - Specifically in relation to this statement, authors must complete the idea they intend to discuss: why would the sample characteristic be determinant for the absence of significant bilateral differences in knee muscle strength?
LINE 316 – I think authors could also expand this discussion (HQ ratio) to ACL injuries, as they are the central theme of the study. What are the normative values?
LINES 343-355 – At this point, I believe it would be interesting to expand the discussion to the sex differences in the use of hip and trunk strategies for postural control and balance, and their relationship to risk factors for ACL injuries.
LINES 356-361 - In the last paragraph, I believe it is necessary to complete the idea and finalize the discussion. [
LIMITATIONS
The authors indicate and/or justify possible limitations of the study, both in relation to the aspect of sample selection, and in relation to the technical and methodological aspects.
The sample size is identified as a limiting factor. A calculation of sample power is indicated to respond it. Can the inclusion and exclusion criteria used also be considered as limitations of the study? Did a self-report without adequate clinical examination (if ocurred) be a limitation?
Only isometric hamstring and quadriceps strengths were tested, and authors must ustify the use of this methodology, since there was an isokinetic dynamometer available
CONCLUSIONS
Authors’ conclusions are based on the study results and point to the association between clinical variables studied.
Reviewer 2 Report
Lines 35-37. Merge these two sentences to improve sentence flow. “Lower-limb injuries are the most common, especially around the knee region [1,2]. In particular, knee sprains, e.g. of the anterior cruciate ligament (ACL), make up around half of all serious knee injuries [3].”
Lines 67-68. The authors have dramatic shift in context from stabilization and asymmetry to discussing body posture absent of a transition. Please add a sentence or two to transition in this sentence and rewrite the sentence so it has improved flow. “However, there are important differences in the body posture and in other boundary conditions between the skiing posture and normal upright standing.” I am difficulty understanding the “boundary conditions” addition to the sentence.
Line 109. Add isokinetic before dynamometer for document consistency.
Lines 115-117. Change the sentence to read something similar to this for better flow, “The testing protocol consisted of two repetitions of isometric maximal voluntary contractions (MVCs) of knee extension and knee flexion for each leg’s muscle groups (quadriceps and hamstring groups).”
Line 309. Is the reason the participants were physically active young adults or because they were experienced skiers?
Line 314. Remove “the” before knee. “To prevent the knee injury and to improve knee joint stability…”
Line 315. Concentric HQ should be assessed in future studies examining dominant and nondominant legs.
Lines 349-352. The sentence needs to be rewritten for clarity as the point trying to be made by the authors is not clear. Additionally, remove anteriorly bent and replace with the joint position of dorsiflexed. The term bent could be interpreted as a physical stress. “Moreover, during wearing ski boots, the ankle joint was slightly anteriorly bent and fixed in the ski boot leading to an ascending shift of the sway axis to the knee joint compared to unipedal barefooted balancing [14,28].”
